# Association between pet ownership and physical function at discharge in hospitalized older adults: A retrospective observational study

Yo Ishihara[1,2,3]*, Kiyomitsu Fukaguchi[2], Hiroshi Koyama[2]

1 Department of Palliative Medicine, International University of Health and Welfare Narita Hospital, Narita, Chiba, Japan, 2 Department of Critical Care Medicine, Shonan Kamakura General Hospital, Kamakura, Kanagawa, Japan, 3 Department of Gastroenterology, International University of Health and Welfare Graduate School of Medicine, Narita, Chiba, Japan

* yo-ishihara@iuhw.ac.jp

## Abstract

### Objectives

Hospitalization can lead to a loss of physical function among older adults. While pet ownership has been reported to have beneficial effects on physical function, its impact on hospitalized patients remains unclear. We examined the association between pet ownership upon admission and physical function during hospitalization.

### Methods

This single-center, retrospective, observational study was conducted between April 2013 and April 2023. Patients aged ≥65 years who were hospitalized for the first time due to pneumonia at our facility were included. Cases were identified through medical record searches using the keywords "pet", "dog", "cat", "ownership", and "keeping". Patients with unclear pet ownership status in the medical records were excluded from the analysis. Patients were classified into two groups: Pet Owners (PO) and Non-Pet Owners (NPO) upon admission. The Barthel index (BI) gain (the difference between admission and discharge BI scores) and BI efficiency (BI gain divided by hospital stay) were compared. Logistic regression was performed for patients with a BI gain >0 and discharge BI scores >85, indicating independence without functional decline, after adjusting for covariates.

### Results

A total of 248 patients were initially screened, we finally included 172 patients (69.4%) in the PO group and 46 patients (18.5%) in the NPO group. The median BI gain was 10 (0–30) and 2.5 (0.0–26.3) in the PO and NPO groups (p = 0.24), respectively. The median BI efficiency was 0.48 (0.00–1.87) and 0.16 (0.00–1.27) in the PO and NPO groups (p = 0.062), respectively. Current pet ownership was not associated

**Data availability statement:** The dataset used in this study includes clinical information that could potentially identify individuals; therefore, it cannot be made publicly available due to ethical constraints. However, the data collected and analyzed in this study may be made available upon reasonable request to Yo Ishihara (yo-ishihara@iuhw.ac.jp) , Kiyomitsu Fukaguchi (fukakiyo1205@gmail.com), and Clinical Trial Center of Shonan Kamakura General Hospital (Email: ccts_kyoyu38@shonankamakura.or.jp), subject to ethical considerations. Data sharing will be considered only if approval is obtained in accordance with relevant ethical guidelines and institutional regulations.

**Funding:** The author(s) received no specific funding for this work.

**Competing interests:** The authors have declared that no competing interests exist.

with independence without functional decline, with an adjusted odds ratio of 0.68 (95% confidence interval, 0.15–3.94; p = 0.63).

## Conclusions

This study found no association between current pet ownership and the maintenance of physical function in hospitalized older adults.

---

## Introduction

Aging is associated with the decline of physical function and reduced quality of life during the end-of-life stage [1]. As the lifespan of older adults continues to increase, maintaining physical function has become more critical [2]. Recent studies suggest that pet ownership may improve health by encouraging physical activity, such as walking with pets [3]. Among community-dwelling older adults aged ≥65 years, those who own pets are less likely to develop frailty than those who never owned pets [4]. Specifically, 12% of dog owners are more active than non-owners. Several studies have proposed that pet ownership positively impacts physical function in older adults [5]. Nevertheless, this advantage may be undermined during hospitalization, as hospital stays significantly reduce physical activity and lead to a marked decline in physical function among older adults [6]. For instance, approximately 18% of patients hospitalized for community-acquired pneumonia have a Barthel index (BI) score of less than 80. A low BI score is associated with longer hospital stays, higher rates of readmission, and increased mortality [7]. BI is an indicator of physical activity, and a clinical tool used to assess the ability to perform basic activities of daily living (ADLs) [8]. BI score below 85 indicates a need for assistance with ADLs [9], while a BI score above 85 suggests the patient is generally independent but may require minimal assistance [10]. It is crucial to maintain and improve physical function, as measured by BI, during and after hospitalization.

Although numerous studies suggest that pet ownership benefits older adults, these studies have focused solely on community-dwelling older adults and have not examined its impact on hospitalized older adults, highlighting a knowledge gap. Whether pet ownership influences the physical functioning of older adults during hospitalization remains unclear. We aimed to explore the relationship between pet ownership and changes in physical function among hospitalized older adults with acute illnesses.

## Methods

### Study design and setting

This single-center retrospective observational study was conducted at Shonan Kamakura General Hospital, a tertiary care hospital in Japan. The study was approved by the Institutional Review Board of the Tokushukai Group Ethics Committee (approval number: TGE02276-024) and conducted in accordance with the

principles outlined in the Declaration of Helsinki. Given the retrospective nature of this study, the requirement for informed consent was waived unless the patient opted out.

## Participants

We included patients aged ≥65 years who were hospitalized for pneumonia between April 2013 and April 2023, and whose medical records at the time of admission contained terms related to pet ownership.

## Data collection

We initially identified electronic medical records of patients whose admission charts contained the following Japanese terms; "pet," "dog," "cat," "ownership," and "keeping," and who were coded as having pneumonia under the insurance disease classification system. As part of the screening process, only the first hospitalization for pneumonia was retained for each patient. Data collected from medical records included patient demographics, pet type (dog, cat, or other), length of hospital stay, BI scores at admission and discharge, ambulance utilization at admission, medical history (hypertension, dyslipidemia, diabetes, chronic heart failure, chronic kidney disease, stroke, malignancy, dementia, schizophrenia, depression, and chronic respiratory diseases, including interstitial pneumonia and chronic obstructive pulmonary disease), as well as whether the patient required mechanical ventilation, vital signs (body temperature, blood pressure, heart rate, respiratory rate, and $SpO_2$), blood pH, $pO_2$, blood glucose levels, hematocrit, serum sodium levels, blood urea nitrogen levels, and the presence or absence of pleural effusion on chest radiography. Based on these variables, the Pneumonia Severity Index (PSI) was calculated for each patient [11]. At our hospital, physical therapists provide rehabilitation and assess the BI for all patients hospitalized with pneumonia upon request of physician. The BI scores at admission and discharge were recorded based on physical therapists' notes in the electronic medical records. All data were collected by a board-certified internist who had completed specialized training in internal medicine.

## Exposures

The primary exposure was pet ownership at the time of hospitalization. Patients were categorized into two groups: Pet Owner (PO) group if they had a pet at the time of admission, and Non-Pet Owner (NPO) group if they did not. Among current pet owners, additional categorizations were made based on the pet type, particularly focusing on dog ownership.

## Outcome measures

The primary outcomes were BI gain and efficiency. BI gain was defined as the difference between BI at admission and discharge, representing functional improvement during hospitalization. BI efficiency is defined as BI gain divided by the length of hospital stay, representing the rate of change in BI gain per hospital day [12]. The secondary outcomes examined the association between current pet ownership and achieving both BI gain >0 and BI at discharge >85, indicating the ability to be discharged independently without a decline in the BI, which is considered a clinically important outcome. Maintaining or improving functional independence is critical for older adults to successfully return to their daily lives and reduce the risk of rehospitalization or long-term care placement. Additionally, BI gain and efficiency were compared between current pet owners who owned dogs and those who owned other types of pets. An overview of this study is shown in Fig 1.

## Confounders

Older age is generally associated with lower physical function and greater difficulty in managing pets. Additionally, men tend to have a shorter average life expectancy compared to women, and women are slightly more likely than men to own pets [4]. Dementia affects both socioeconomic status and physical function, and socioeconomic status influences current

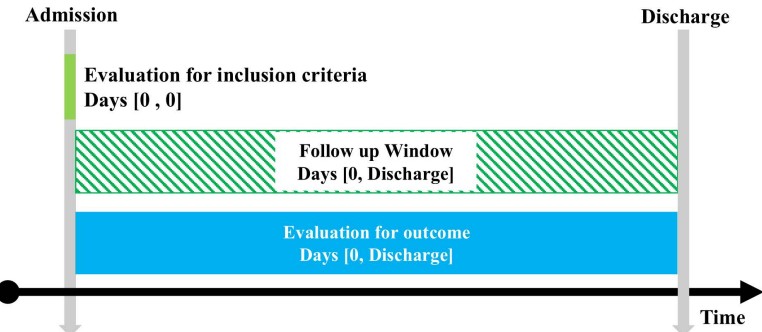

**Fig 1. Design diagram.** The study design diagram is shown.

pet-ownership [4]. Thus, age, sex, and dementia status were considered confounding factors in the association between pet ownership and BI. Clinically, pet ownership is unlikely to influence the severity of pneumonia; however, the severity of pneumonia has a substantial impact on physical function. Although pneumonia severity, as reflected by PSI, is unlikely to be affected by pet ownership, it may significantly influence physical function outcomes. PSI may act as a prognostic factor, so we included the PSI as an adjustment variable to examine the direct effect of pet ownership on physical function. We sought the controlled direct effect of pet ownership on discharge BI, conditional on PSI. Given the Directed Acyclic Graph assumptions (S1 Fig), PSI lies outside the causal pathway from exposure to outcome; hence conditioning on it is expected to improve precision without inducing collider bias. For completeness, we also report an unadjusted-for-PSI model (S1 Table) to illustrate the total effect. PSI was classified into Classes I–V based on the original criteria, with Classes IV and V indicating severe cases.

## Statistical analysis

Comparisons between the PO and NPO groups were performed as follows: Categorical variables were analyzed using the chi-squared test, while continuous variables were compared by calculating the median and performing the Wilcoxon rank-sum test. For continuous variables, the median and interquartile range (IQR) were reported. Logistic regression analysis was used to assess the relationship between pet ownership and both BI gain >0 during hospitalization and BI > 85 at discharge, with age, sex, dementia, and PSI included as covariates. Odds ratios (ORs) and 95% confidence interval (CI) were calculated. For subgroup analysis among pet owners, the median values of BI gain and BI efficiency were compared between dog owners and non-dog owners using the Wilcoxon rank-sum test. Logistic regression analysis was conducted to assess the association between dog ownership and both BI gain >0 during hospitalization and BI > 85 at discharge, adjusting for age, sex, dementia, and PSI. All statistical tests were two-sided, with p < 0.05 considered statistically significant. Statistical analyses were performed using R (version 4.4.3, R Foundation for Statistical Computing, Vienna, Austria).

## Results

### Patient characteristics

Fig 2 presents a CONSORT diagram of the study. A total of 248 patients were initially identified. After excluding patients with unclear documentation of pet ownership, we finally included 172 patients in the PO group and 46 patients in the NPO group.

Table 1 presents the characteristics of the patients. The median age (IQR) was 78 (70.8–84.0) and 80 (74–85) years in the PO and NPO groups, respectively. There were 116 (67.4%) and 35 (76.1%) male in the PO and NPO groups, respectively. In the PO group, 110 patients (64.0%) owned dogs only, 59 (34.3%) owned cats only, 13 (7.6%) owned

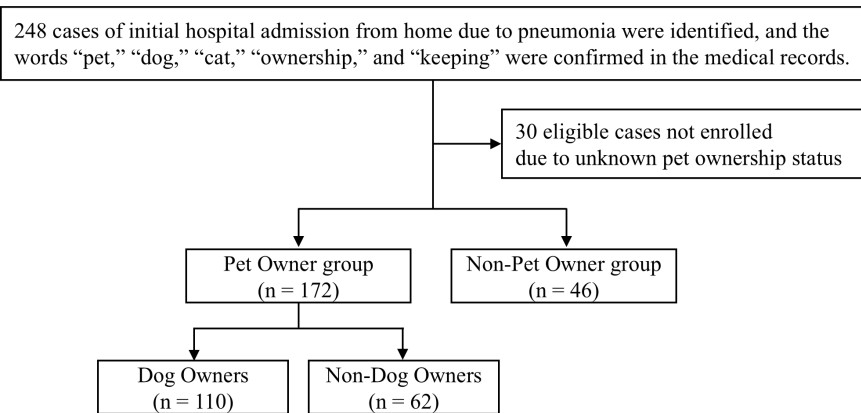

**Fig 2. CONSORT diagram.** Flow of participants through the study.

both dogs and cats, and 20 (11.6%) owned other types of pets. Other pets included Japanese rice fish (medaka), ferrets, turtles, goldfish, birds (such as parakeets, Java sparrows, finches, and pigeons), rabbits, hamsters, squirrels, guinea pigs, and armadillo. At admission, the median BI (IQR) was 50 (16.3–80.0) and 42.5 (30.0–78.8) in the PO and NPO groups, respectively. At discharge, the median BI (IQR) was 80 (50–85) and 65 (45–85) in the PO and NPO groups, respectively.

## Outcome evaluation

The median BI gain (IQR) was 10 (0–30) and 2.5 (0.0–26.3) in the PO and NPO groups, respectively, with no significant difference (p = 0.24). The median BI efficiency (IQR) was 0.48 (0.001.87) and 0.16 (0.00–1.27) in the PO and NPO groups, respectively, with no significant difference (p = 0.062) (Table 2).

In the multivariable logistic regression analysis, current pet ownership was not significantly associated with both a BI gain of >0 and a discharge BI of >85, with an adjusted ORs of 0.68 (95% CI, 0.15–3.94; p = 0.63) (Table 3). There were no missing values in the covariates.

Among current pet owners, the median BI gain (IQR) was 10 (0–30) and 7.5 (0–30) for dog and non-dog owners, respectively (p = 0.97). The median BI efficiency (IQR) was 0.60 (0.00–1.83) and 0.37 (0.00–2.23) for dog owners and owners of other pet types, respectively (p = 0.76) (Table 4).

Dog ownership was not significantly associated with both a BI gain of >0 and a discharge BI of >85, with an adjusted ORs of 0.68 (95% CI, 0.15–3.94; p = 0.63) (Table 5). There were no missing values in the covariates.

## Discussion

This study found no evidence of an association between current pet ownership and either physical function or changes in BI scores among older adults hospitalized with pneumonia. Even after adjusting for age, sex, dementia, and PSI more than class IV as covariates, pet ownership was not associated with improved BI scores and functional independence at discharge. To date, no other studies have examined the relationship between pet ownership and physical health during hospitalization. To our knowledge, this is the first study to investigate whether pet ownership influences physical function during hospitalization and whether dog ownership has a greater impact than ownership of other types of pets.

Older adults often experience a decline in physical activity levels following hospitalization due to bed, either as a result of disease-related exhaustion or iatrogenic conditions. Even a single day of hospitalization can increase sedentary time by at least 8 minutes in outpatients [13]. Therefore, increasing physical activity and reducing sedentary time during hospitalization are crucial [14]. Several studies have explored physical function maintenance in older

**Table 1. Characteristics of patients hospitalized with pneumonia.**

| | Pet Owner group (n = 172) | Non-Pet Owner group (n = 46) |
|---|---|---|
| **Characteristic** | | |
| Male, n (%) | 116 (67.4) | 35 (76.1) |
| Age, median (IQR) | 78 (70.8-84.0) | 80 (74-85) |
| Length of hospital stay, median (IQR) | 12 (7-28) | 20 (10.5-30.8) |
| Transported by Ambulance, n (%) | 87 (50.6) | 22 (47.8) |
| **Barthel index (BI)** | | |
| BI on admission (IQR)[a] | 50 (16.3-80.0) | 42.5 (30.0-78.8) |
| BI on discharge (IQR)[a] | 80 (50-85) | 65 (45-85) |
| BI gain >0 and discharge BI >85, n (%) | 11 (6.4) | 2 (4.3) |
| **Type of pneumonia** | | |
| Community associated pneumoniae, n (%) | 126 (73.3) | 18 (39.1) |
| Interstitial pneumoniae, n (%) | 41 (23.8) | 28 (60.9) |
| Eosinophilic pneumoniae, n (%) | 3 (1.7) | 0 (0.0) |
| Hypersensitivity pneumoniae, n (%) | 2 (1.2) | 0 (0.0) |
| **Type of pet animals** | | |
| Dog ownership, n (%) | 110 (64.0) | – |
| Cat ownership, n (%) | 59 (34.3) | – |
| Dog and cat ownership, n (%) | 13 (7.6) | – |
| Other pets ownership, n (%) | 20 (11.6) | – |
| **Past medical history** | | |
| Hypertension, n (%) | 88 (51.2) | 24 (52.2) |
| Hyperlipidemia, n (%) | 44 (25.6) | 11 (23.9) |
| Diabetes, n (%) | 29 (16.9) | 12 (26.1) |
| Stroke, n (%) | 23 (13.4) | 5 (10.9) |
| Chronic Heart Failure, n (%) | 17 (9.9) | 3 (6.5) |
| Chronic Kidney Disease, n (%) | 17 (9.9) | 3 (6.5) |
| Malignancy, n (%) | 31 (18.0) | 8 (17.4) |
| Dementia, n (%) | 17 (9.9) | 3 (6.5) |
| Schizophrenia, n (%) | 2 (1.2) | 0 (0.0) |
| Depression, n (%) | 2 (1.2) | 0 (0.0) |
| Chronic respiratory disease, n (%) | 61 (35.5) | 20 (43.5) |
| **Pneumonia Severe Index Risk class** | | |
| I | 0 (0.0) | 0 (0.0) |
| II | 6 (3.5) | 2 (4.3) |
| III | 39 (22.7) | 8 (17.4) |
| IV | 75 (43.6) | 16 (34.8) |
| V | 52 (30.2) | 19 (41.3) |
| **Mechanical ventilation** | 10 (5.8) | 2 (4.3) |
| **In-hospital death, n (%)** | 21 (12.2) | 11 (23.9) |

IQR, interquartile range.

[a]The BI was calculated only for patients with available rehabilitation records: 129 (88.4%) in the PO group and 39 (84.8%) in the NPO group.

**Table 2. Comparison of Barthel index gain and efficiency between pet owner group and non-pet owner group.**

|  | Pet Owner group | Non-Pet Owner group | *p* value |
|---|---|---|---|
|  | (n = 172) | (n = 46) |  |
| Barthel index gain, median (IQR) | 10 (0-30) | 2.5 (0.0-26.3) | 0.24 |
| Barthel index efficiency, median (IQR) | 0.48 (0.00-1.87) | 0.16 (0.00-1.27) | 0.062 |

IQR, interquartile range

**Table 3. Logistic regression analysis of Barthel index gain of >0 and Barthel index at discharge of >85 according to pet ownership.**

|  | Crude odds ratio (95% confidence interval) | Adjusted odds ratio (95% confidence interval) | *p* value |
|---|---|---|---|
| Pet ownership | 1.47 (0.37-9.91) | 0.68 (0.15-3.94) | 0.63 |

**Table 4. Comparison of Barthel index gain and efficiency between dog owner and non-dog owner in pet owner group.**

|  | Pet ownership | | *p* value |
|---|---|---|---|
|  | Dog owner | Non-dog owner |  |
|  | (n = 110) | (n = 62) |  |
| Barthel index gain, median (IQR) | 10 (0-30) | 7.5 (0-30) | 0.97 |
| Barthel index efficiency, median (IQR) | 0.60 (0.00-1.83) | 0.37 (0.00-2.23) | 0.76 |

IQR, interquartile range

**Table 5. Logistic regression analysis of Barthel index gain of >0 and Barthel index at discharge of >85 according to dog ownership.**

|  | Crude odds ratio (95% confidence interval) | Adjusted odds ratio (95% confidence interval) | p value |
|---|---|---|---|
| Dog ownership | 1.47 (0.37-9.91) | 0.68 (0.15-3.94) | 0.63 |

adults, with many suggested that pet ownership may promote better community health by increasing physical activity [4,15–23]. However, in the present study, no significant differences were observed in the degree of physical function improvement between the PO and NPO groups. There are several possible explanations for this finding. First, the NPO group included individuals with a history of pet ownership and those who had never owned pets. Previous studies have reported that individuals with a history of pet ownership, regardless of whether they owned a dog or cat, had significantly lower odds of developing new-onset frailty compared to those with no history of pet ownership. In other words, prior pet ownership is associated with a reduced risk of frailty [4]. This suggests that some individuals in the NPO group may have had a lower baseline risk of developing frailty. Second, the participants were unable to interact directly with animals during hospitalization. A randomized controlled trial conducted in a long-term care facility found a trend toward increasing BI scores over time among residents who interacted with dogs twice a week [24]. The absence of direct contact with animals during hospitalization may have negated the usual benefits of pet ownership, particularly for individuals in the PO group. However, in both the comparison of BI gain and efficiency and the regression analyses, the point estimates favored the PO group, suggesting the study may have been underpowered due to its limited sample size.

The impact of pet ownership on physical activity varies by animal species. For instance, dog owners are more likely than non-dog owners to engage in at least 150 minutes of physical activity per week [3]. Dog owners also walk more frequently than non-pet owners, whereas cat ownership has no significant impact on health or walking behavior

[17]. Additionally, cat owners tend to be less physically active and more sedentary than dog owners [15]. Other pets, such as cats, birds, and small mammals, are not associated with high levels of physical activity [25]. Given that dog ownership has been shown to have a positive impact on physical activity, it is important to differentiate between dog and non-dog pet owners when evaluating the effects of pet ownership on physical activity and overall health. Accordingly, the present study specifically examined dog ownership among pet owners. However, no significant differences were observed in the degree or efficiency of physical function improvement during hospitalization between dog owners and non-dog owners. Previous studies indicate that approximately 70% of dog owners walk their dogs, with nearly 90% of them walking their dogs one to two times per day. Approximately half of these owners walk for more than 30 min per session [3]. The lack of a dog-walking routine due to hospitalization may have influenced the findings of the present study.

While we aim to further investigate the effects of animals on physical function in older adults, the present study has some limitations. First, the ability to collect detailed medical records was limited and patient selection relied on keyword searches, which may have introduced bias. Because cases that did not match the keyword search criteria were not extracted, there is a possibility of selection bias. The cases which could not be included in the analysis due to the absence of pet ownership information, may have further increased selection bias. However, particular attention was paid to using keywords that were considered sufficiently appropriate for capturing the relevant cases. Furthermore, for all cases identified through the keyword search, detailed information was carefully collected by manually reviewing the electronic medical records. At our institution, pet ownership history is not routinely recorded for conditions other than pneumonia, such as urinary tract infections. However, in cases of pneumonia, information on animal contact is often documented during admission interviews, particularly with consideration of zoonotic diseases such as psittacosis. Therefore, to minimize selection bias, we focused on patients with pneumonia, for whom physicians were encouraged to record pet ownership at the time of admission. Second, the sample size was relatively small due to this patient selection method. Third, some confounding factors were not accounted for, including psychological state, presence of supportive individuals in the patient's household, and levels of social participation and employment, all of which can influence pet ownership and physical activity. Socioeconomic status, baseline physical activity, area of residence, and presence of other residents could theoretically act as confounders, as they may influence both pet ownership and physical function on DAG. However, these variables were not consistently documented in the electronic medical records and could not be reliably incorporated into the analysis. Proxy information for these variables was also not available in the electronic medical records. Furthermore, the severity of stroke or malignancy may have contributed to a lower baseline physical function. As this was an observational study based solely on electronic medical record data, information on social participation, employment status, psychological condition, and physical impairments could not be collected unless documented, and such details were rarely available. This represents a major limitation of this study. To gain a more comprehensive understanding of the impact of pet ownership on daily life, future studies should incorporate surveys or questionnaires, conduct prospective research, and include a broader patient population regardless of pneumonia diagnosis.

In conclusion, the present study found no evidence that pet ownership at home is associated with improved physical function during hospitalization or with independence in ADLs at discharge among older adults hospitalized for pneumonia. However, several limitations, including potential selection bias, may have influenced the results. Further research is required to clarify the extent of the impact of pet ownership and explore methods to further enhance physical function in hospitalized older adults to improve their quality of life.

## Supporting information

**S1 Fig. Directed Acyclic Graph.** Directed Acyclic Graph of the study.
(TIF)

**S1 Table. Logistic regression analysis without adjustment for Pneumonia Severity Index (PSI).** This model illustrates the total effect of pet ownership on the outcome, without conditioning on PSI, which lies outside the causal pathway according to the assumptions of the Directed Acyclic Graph.
(DOCX)

## Acknowledgement

We thank Mr. Ryota Hara for helping with data collection.

## Author contributions

**Conceptualization:** Yo Ishihara.

**Data curation:** Yo Ishihara.

**Formal analysis:** Yo Ishihara.

**Investigation:** Yo Ishihara.

**Methodology:** Yo Ishihara, Kiyomitsu Fukaguchi.

**Project administration:** Yo Ishihara.

**Supervision:** Kiyomitsu Fukaguchi, Hiroshi Koyama.

**Visualization:** Yo Ishihara.

**Writing – original draft:** Yo Ishihara.

**Writing – review & editing:** Yo Ishihara, Kiyomitsu Fukaguchi, Hiroshi Koyama.

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
