## [Decision Letter · Decision Letter 0]

23 Mar 2025

Dear Dr. Ishihara,

Thank you for submitting your manuscript to PLOS ONE. After careful consideration, we feel that it has merit but does not fully meet PLOS ONE’s publication criteria as it currently stands. Therefore, we invite you to submit a revised version of the manuscript that addresses the points raised during the review process.

We look forward to receiving your revised manuscript.

Kind regards,

Amr Ehab El-Qushayri

Academic Editor

PLOS ONE

Additional Editor Comments:

I would congratulate the authors for their paper. I had few comments in addition to the reviewers' comments:

1-Please indicate the grade of pneumonia (mild, moderate or severe) since it can affect the level of physical activity.

2-Please indicate the method of pneumonia treatment as it also affect the level of physical activity. Did any patient receive mechanical ventilation,...etc

3-There is a major limitation here that could turn your results to be insignificant which is the comorbidity and the elder age. Many patients had stroke, malignancy, even severe cases of pneumonia,..etc. Those patients would have low physical activity whether they own pets or no. Please elaborate in discussion and also explain this in conclusion for future studies as this is a strong limitation in your study.

4-Rewrite the conclusion since it is firm while you had many limitations that would turn the results upside down if a new study have addressed them well.

Reviewers' comments:

Reviewer's Responses to Questions

**Comments to the Author**

1. Is the manuscript technically sound, and do the data support the conclusions?

Reviewer #1: Yes

Reviewer #2: Partly

2. Has the statistical analysis been performed appropriately and rigorously?

Reviewer #1: Yes

Reviewer #2: Yes

3. Have the authors made all data underlying the findings in their manuscript fully available?

Reviewer #1: Yes

Reviewer #2: No

4. Is the manuscript presented in an intelligible fashion and written in standard English?

Reviewer #1: Yes

Reviewer #2: Yes

Reviewer #1: Clarity: The article is well-written, with concise and coherent language.

Argument Strength: While arguments are clear, the idea lacks strong significance.

Research Quality: It features robust statistics, solid methodologies, and credible sources.

Limitations: Thoughtful limitations are acknowledged, enhancing credibility.

Relevance: The topic holds less importance due to existing hospital rules about pets.

Structure: The introduction, results, and conclusion are logically and effectively organized.

Style and Tone: The language is appropriate and tailored for a diverse audience

Reviewer #2: The study has a good methodological design, particularly for its use of Directed Acyclic Graphs (DAGs) and a detailed flowchart outlining the study process. Incorporating DAGs to visually represent assumptions about causal relationships enhances the transparency and rigor of the analysis, allowing readers to clearly understand the rationale behind variable selection and adjustment strategies. Additionally, the inclusion of a study flowchart provides valuable clarity regarding participant selection, exclusions, and key study stages. However, these are my comments

Introduction:

Please add more information about Barthel index

Methods:

90-97

The authors report that they searched electronic medical records using Japanese terms such as “pet,” “dog,” “cat,” “ownership,” and “keeping,” in combination with the insurance disease name “pneumonia” to identify eligible patients. However, this approach raises some concerns regarding the reliability and reproducibility of case identification.

Specifically, Japan employs a standardized coding system (e.g., ICD-10 codes) in its national health insurance claims data, which should provide a more accurate and systematic method for identifying cases of pneumonia. It is unclear why the authors chose to rely on keyword searches in the free text or insurance disease name fields rather than using established diagnostic codes.

Using text-based searches may increase the risk of selection bias or misclassification, particularly if terminology varies among physicians. Moreover, this method could miss cases that are coded appropriately but do not contain the exact keywords used, or could include cases that mention the keywords without confirming a true diagnosis.

To enhance methodological rigor and reproducibility, the authors should justify this choice and, if possible, demonstrate the validity of their text-based approach compared to standard coding methods.

Moreover, BI scores were recorded based on physical therapists’ notes, but it does not mention whether these assessments were standardized or validated across therapists. This raises concerns about inter-rater reliability.

99-106: Why the author choose specific comorbidities and omit others like chronic kidney disease or heart failure, Also, psychiatric disease should be specified as this term is too broad (depression, anxiety and so on)

135-142

The DAG may omit key confounders that could affect both pet ownership and physical function. For example, socioeconomic status, baseline physical activity, or cognitive function might influence both exposure and outcome.

Additionally, as a reader, I would appreciate a clearer explanation of the rationale behind the selection of adjustment variables. For example, it is unclear why psychological status was not included—was it considered a collider or irrelevant in the causal pathway? Including this type of justification would strengthen the methodological transparency. It would also be helpful if the DAGs were labeled to indicate the role of each variable (e.g., confounder, mediator, collider), which would make it easier to understand the assumptions guiding the adjustment strategy.

144-156

It is important to note that gender and sex represent distinct concepts; in the DAGs, gender was depicted, while the Methods section reported on sex.

Results:

196

While the authors present main regression outcome in the results section, they do not provide the complete regression table, including all covariates, their coefficients, confidence intervals, and p-values.

Discussion

213-219

252-254: why it is difficult to include those variables ?

Figure 2 shows that 62 patients were excluded from the analysis due to the absence of rehabilitation records. While this criterion is understandable for ensuring data completeness, such an exclusion could introduce selection bias if the lack of rehabilitation records is not random. For example, patients without rehabilitation records may differ systematically from those with records in terms of disease severity, functional status, or access to care. This could potentially bias the results and affect the generalizability of the findings.

The authors should acknowledge this as a limitation in the discussion section. Additionally, they could provide a brief comparison of baseline characteristics between included and excluded patients to assess the potential for bias.

**Do you want your identity to be public for this peer review?** For information about this choice, including consent withdrawal, please see our Privacy Policy

Reviewer #1: No

Reviewer #2: No

---

## [Author Response · Author response to Decision Letter 1]

27 May 2025

Thank you for your ongoing consideration of our manuscript for publication in PLOS One. We are grateful for the time and effort you and the reviewers have dedicated to this process. We believe the revised manuscript has been improved. Below, we have addressed the comments provided by you and the reviewers.

We look forward to your editorial decision.

Sincerely,

Yo Ishihara, MD

Email: yo-ishihara@iuhw.ac.jp

Editor:

1. Please indicate the grade of pneumonia (mild, moderate or severe) since it can affect the level of physical activity.

[Response]

Thank you for your valuable comment. We have re-examined the electronic records of all patients, reassigned the Pneumonia Severity Index (PSI) scores, and presented the distribution by class I–V in the revised Table 1. Furthermore, in the logistic regression analysis, we have changed the adjustment factor for severity from ambulance use to PSI.

2. Please indicate the method of pneumonia treatment as it also affect the level of physical activity. Did any patient receive mechanical ventilation,...etc

[Response]

Thank you for your comment. We have summarized the data on patients who received mechanical ventilation and included it in the revised Table 1.

3. There is a major limitation here that could turn your results to be insignificant which is the comorbidity and the elder age. Many patients had stroke, malignancy, even severe cases of pneumonia,..etc. Those patients would have low physical activity whether they own pets or no. Please elaborate in discussion and also explain this in conclusion for future studies as this is a strong limitation in your study.

[Response]

Thank you for your comment. We have added the following statement to the Discussion section of the revised manuscript on Lines 270-275. “Furthermore, the severity of stroke or malignancy may have contributed to a lower baseline physical function. As this was an observational study based solely on electronic medical record data, information on social participation, employment status, psychological condition, and physical impairments could not be collected unless documented, and such details were rarely available. This represents a major limitation of this study.”

4. Rewrite the conclusion since it is firm while you had many limitations that would turn the results upside down if a new study have addressed them well.

[Response]

Thank you for your comment. We have revised the conclusion of the study as follows.

Lines 279-285: “In conclusion, the present study found no evidence that pet ownership at home is associated with improved physical function during hospitalization or greater independence in ADLs at discharge among older adults hospitalized for pneumonia. However, several limitations, including potential selection bias, may have influenced the results. Further research is required to clarify the extent of the impact of pet ownership and explore methods to further enhance physical function in hospitalized older adults to improve their quality of life.”

Reviewer: 2

1. INTRODUCTION: Please add more information about Barthel index

[Response]

Thank you for your valuable suggestions. We have added information about the Barthel index on lines 66-71 in the Introduction section as follows: “BI is an indicator of physical activity, and a clinical tool used to assess the ability to perform basic activities of daily living (ADLs) [8]. BI score below 85 indicates a need for assistance with ADLs [9], while a BI score above 85 suggests the patient is generally independent but may require minimal assistance [10].”

2. METHOD: The authors report that they searched electronic medical records using Japanese terms such as “pet,” “dog,” “cat,” “ownership,” and “keeping,” in combination with the insurance disease name “pneumonia” to identify eligible patients. However, this approach raises some concerns regarding the reliability and reproducibility of case identification. Specifically, Japan employs a standardized coding system (e.g., ICD-10 codes) in its national health insurance claims data, which should provide a more accurate and systematic method for identifying cases of pneumonia. It is unclear why the authors chose to rely on keyword searches in the free text or insurance disease name fields rather than using established diagnostic codes. Using text-based searches may increase the risk of selection bias or misclassification, particularly if terminology varies among physicians. Moreover, this method could miss cases that are coded appropriately but do not contain the exact keywords used, or could include cases that mention the keywords without confirming a true diagnosis. To enhance methodological rigor and reproducibility, the authors should justify this choice and, if possible, demonstrate the validity of their text-based approach compared to standard coding methods. Moreover, BI scores were recorded based on physical therapists’ notes, but it does not mention whether these assessments were standardized or validated across therapists. This raises concerns about inter-rater reliability.

[Response]

Thank you for your valuable comment. Patients were included if pneumonia was detected as the principal diagnosis, the diagnosis leading to hospitalization, or the diagnosis requiring the greatest medical resources. For these cases, we reviewed all inpatient electronic medical records, including imaging findings, and board-certified physicians who had completed specialized training in internal medicine assessed whether the diagnosis of pneumonia was appropriate. Therefore, cases in which pneumonia was not truly diagnosed despite the presence of keywords were not included in the study. Conversely, there was a possibility of missing cases that did not contain the specified keywords, and we recognize this as a limitation of our study. It was not feasible to collect information regarding pet ownership through methods other than keyword searches within the hospital’s electronic medical record system. BI scores were recorded based on the notes written by physical therapists in the electronic medical records. Although it is extremely difficult to verify whether the assessments were standardized across different evaluators based on the available records, it is certain that all rehabilitation interventions were performed by licensed physical therapists and that the BI scores were documented based on their professional knowledge and experience.

2. METHOD: 99-106: Why the author choose specific comorbidities and omit others like chronic kidney disease or heart failure, Also, psychiatric disease should be specified as this term is too broad (depression, anxiety and so on)

[Response]

Thank you for your valuable comment. We have reclassified the comorbidities as dementia, schizophrenia, and depression and revised the Methods section accordingly. As you pointed out, chronic kidney disease and heart failure are also likely to affect ADLs; therefore, we have added these comorbidities as well. All of these variables have been included in the revised Table 1.

3. METHOD: The DAG may omit key confounders that could affect both pet ownership and physical function. For example, socioeconomic status, baseline physical activity, or cognitive function might influence both exposure and outcome. Additionally, as a reader, I would appreciate a clearer explanation of the rationale behind the selection of adjustment variables. For example, it is unclear why psychological status was not included—was it considered a collider or irrelevant in the causal pathway? Including this type of justification would strengthen the methodological transparency. It would also be helpful if the DAGs were labeled to indicate the role of each variable (e.g., confounder, mediator, collider), which would make it easier to understand the assumptions guiding the adjustment strategy.

[Response]

Thank you for your valuable comment. We have revised the DAG by adding markers to clarify the roles of confounders and mediators. We have also explicitly described the rationale for selecting confounders in the Methods section of the revised manuscript. Regarding psychological status, it was not included because information on psychological conditions could not be obtained from the electronic medical records.

4. METHOD: 144-156: It is important to note that gender and sex represent distinct concepts; in the DAGs, gender was depicted, while the Methods section reported on sex.

[Response]

Thank you for your suggestions. We have revised the DAG by replacing “gender” with “sex.”

5. RESULTS: 196: While the authors present main regression outcome in the results section, they do not provide the complete regression table, including all covariates, their coefficients, confidence intervals, and p-values.

[Response]

Thank you for your valuable comment. We have now provided the regression coefficients, confidence intervals, and p-values for all covariates in Table 3.

6�DISCUSSION: 213-219: 252-254: why it is difficult to include those variables ?　Figure 2 shows that 62 patients were excluded from the analysis due to the absence of rehabilitation records. While this criterion is understandable for ensuring data completeness, such an exclusion could introduce selection bias if the lack of rehabilitation records is not random. For example, patients without rehabilitation records may differ systematically from those with records in terms of disease severity, functional status, or access to care. This could potentially bias the results and affect the generalizability of the findings. The authors should acknowledge this as a limitation in the discussion section. Additionally, they could provide a brief comparison of baseline characteristics between included and excluded patients to assess the potential for bias.

[Response]

Thank you for your valuable comment. Because the absence of rehabilitation records prevented evaluation of the outcome, patients without rehabilitation records were excluded from the study. After reanalyzing the data, we found that 59 patients lacked rehabilitation records. Figure 1 has been revised accordingly. The absence of rehabilitation records likely indicates that no rehabilitation referral was made by the attending physician; however, whether this process was random could not be determined. Given the potential for selection bias, we compared the characteristics of patients without rehabilitation records and those who participated in the study, as presented in the S2 Table. The results showed that patients without rehabilitation records had a significantly lower proportion of men and were significantly younger. Additionally, the prevalence of prior stroke was significantly lower among patients without rehabilitation records, whereas no significant differences were observed between the groups regarding other comorbidities or pneumonia severity. As age, sex, and history of stroke are factors that can influence rehabilitation outcomes and prognosis, these findings suggest the presence of selection bias in the present study.

---

## [Decision Letter · Decision Letter 1]

15 Jun 2025

Dear Dr. Ishihara,

Thank you for submitting your manuscript to PLOS ONE. After careful consideration, we feel that it has merit but does not fully meet PLOS ONE’s publication criteria as it currently stands. Therefore, we invite you to submit a revised version of the manuscript that addresses the points raised during the review process.

We look forward to receiving your revised manuscript.

Kind regards,

Amr Ehab El-Qushayri

Academic Editor

PLOS ONE

Journal Requirements:

Reviewers' comments:

Reviewer's Responses to Questions

**Comments to the Author**

Reviewer #2: All comments have been addressed

Reviewer #3: (No Response)

2. Is the manuscript technically sound, and do the data support the conclusions?

Reviewer #2: Yes

Reviewer #3: Yes

3. Has the statistical analysis been performed appropriately and rigorously?

Reviewer #2: No

Reviewer #3: No

4. Have the authors made all data underlying the findings in their manuscript fully available?

Reviewer #2: Yes

Reviewer #3: No

5. Is the manuscript presented in an intelligible fashion and written in standard English?

Reviewer #2: Yes

Reviewer #3: Yes

Reviewer #2: For comment "METHOD: The DAG may omit key confounders that could affect both pet

ownership and physical function. For example, socioeconomic status, baseline

physical activity, or cognitive function might influence both exposure and outcome.

Additionally, as a reader, I would appreciate a clearer explanation of the rationale

behind the selection of adjustment variables. For example, it is unclear why

psychological status was not included—was it considered a collider or irrelevant in

the causal pathway? Including this type of justification would strengthen the

methodological transparency. It would also be helpful if the DAGs were labeled to

indicate the role of each variable (e.g., confounder, mediator, collider), which would

make it easier to understand the assumptions guiding the adjustment strategy."

While I appreciate the inclusion of labeled DAGs and partial rationale in the Methods section, the justification for selecting only three adjustment variables remains insufficient. Psychological status was excluded due to lack of data, rather than a theoretical justification based on the causal framework. Additionally, including the Pneumonia Severity Index (PSI) as an adjustment variable—despite acknowledging it may act as a mediator—raises concerns about potential bias from conditioning on a variable in the causal pathway. The rationale for estimating a direct versus total effect is not discussed. I encourage the authors to:

Justify the inclusion of PSI under this estimand;

More comprehensively discuss why other plausible confounders (e.g., SES, baseline physical activity, cognitive function) were excluded;

If model complexity was a constraint, please report the sample size and modeling approach used to evaluate overfitting.

Reviewer #3: lines 139-140: you mentioned that Pneumonia severity may act as a mediator and a previous statement is that Pet ownership is unlikely to influence the severity of Pneumonia. A mediator should be in the middle of the pathway between ethe exposure and the outcome. If Pneumonia was not caused by pet ownership it cant be a mediator. However, you can still adjust for it if it causes the outcome; so no need to change the analysis.

Fig1: any Inclusion or exclusion that happens on future events after the exposure can cause "Immortal time Bias"; lacking records of pet ownership and being hospitalized for the first time happens at the admission. Records missing should be treated as data missing not as an exclusion criteria. Patients that died during hospitalization should still be included and then censored at death. Please update the graph and the analysis accordingly. The graph should have no conditioning on future events (i.e. no inclusion, or exclusion criteria on the follow up period), and the characteristics table (the analysis) should include patients who owns pets upon admission, later on, if death occurs, they may be censored. Meaning they should be added to the baseline variables calculations, and then censored at the outcome.

**Do you want your identity to be public for this peer review?** For information about this choice, including consent withdrawal, please see our Privacy Policy

Reviewer #2: No

Reviewer #3: **Yes: ** Ahmed Mostafa Ahmed Kamel

---

## [Author Response · Author response to Decision Letter 2]

18 Jul 2025

Revisions

1. We made minor revisions to the English expressions throughout the manuscript.

2. The affiliation of the first author has been partially revised.

3. We revised the Participants section as follows: “We included patients aged ≥65 years who were hospitalized for pneumonia and whose medical records at the time of admission contained terms related to pet ownership.”(Lines 84-86)

4. We added the following sentence to the Data Collection section: “We initially identified electronic medical records of patients whose admission charts contained the following Japanese terms; “pet,” “dog,” “cat,” “ownership,” and “keeping,” and who were coded with a diagnosis of pneumonia under the insurance disease classification system.” (Lines 89-92)

5. We added “Dementia affects both socioeconomic status and physical function, and socioeconomic status influences current pet-ownership.” (Lines 133-135)

6. We added “Other pets included Japanese rice fish (medaka), ferrets, turtles, goldfish, birds (such as parakeets, Java sparrows, finches, and pigeons), rabbits, hamsters, squirrels, guinea pigs, and armadillo,” because the revised exclusion criteria resulted in an increased number of eligible patients. (Lines 177-179)

7. We added “Even after adjusting for age, sex, dementia, and PSI more than class IV as covariates, pet ownership was not associated with improved BI scores and functional independence at discharge.” (Lines 212-214)

8. We also revised the results in Table 3. Since our primary aim was to estimate the causal effect of pet ownership on Barthel Index at discharge, we believe that the odds ratios and confidence intervals of other covariates are not interpretable and thus do not warrant inclusion. (Table 2)

9. The order of References [11] and [12] has been corrected to match their order of first citation in the manuscript.

10. We revised the Exposure section as follows: “The primary exposure was pet ownership at the time of hospitalization. Patients were categorized into two groups: Pet Owner (PO) group if they had a pet at the time of admission, and Non-Pet Owner (NPO) group if they did not. Among current pet owners, additional categorizations were made based on the pet type, particularly focusing on dog ownership.” (Lines 110-113)

11. We added “Socioeconomic status, baseline physical activity, and cognitive function could theoretically act as confounders, as they may influence both pet ownership and physical function on DAG. However, these variables were not consistently documented in the electronic medical records and could not be reliably incorporated into the analysis. Proxy information for these variables was also not available in the electronic medical records.” (Lines 275-280)

12. We added logistic regression analysis of BI gain>0 and BI index at discharge>85 according to Dog ownership. (Table 5)

Journal Requirements:

[Response]

We have removed the previous reference:

12. W L, Vde M, RL, WB, NK, GT, Town GI, et al. Defining community acquired pneumonia severity on presentation to hospital: An international derivation and validation study. Thorax. 2003;58: 377–382. doi: 10.1136/thorax.58.5.377.

and replaced it with the following reference:

12. Fine MJ, Auble TE, Yealy DM, Hanusa BH, Weissfeld LA, Singer DE, et al. A prediction rule to identify low-risk patients with community-acquired pneumonia. N Engl J Med. 1997;336(4):243–250. doi: 10.1056/NEJM199701233360402.

This change was made because we re-collected the data using the Pneumonia Severity Index (PSI) instead of the CURB-65 score to assess pneumonia severity in our study.

Reviewer: 2

For comment "METHOD: The DAG may omit key confounders that could affect both pet ownership and physical function. For example, socioeconomic status, baseline physical activity, or cognitive function might influence both exposure and outcome. Additionally, as a reader, I would appreciate a clearer explanation of the rationale behind the selection of adjustment variables. For example, it is unclear why psychological status was not included—was it considered a collider or irrelevant in the causal pathway? Including this type of justification would strengthen the

methodological transparency. It would also be helpful if the DAGs were labeled to

indicate the role of each variable (e.g., confounder, mediator, collider), which would

make it easier to understand the assumptions guiding the adjustment strategy."

While I appreciate the inclusion of labeled DAGs and partial rationale in the Methods section, the justification for selecting only three adjustment variables remains insufficient. Psychological status was excluded due to lack of data, rather than a theoretical justification based on the causal framework. Additionally, including the Pneumonia Severity Index (PSI) as an adjustment variable—despite acknowledging it may act as a mediator—raises concerns about potential bias from conditioning on a variable in the causal pathway. The rationale for estimating a direct versus total effect is not discussed. I encourage the authors to:

Justify the inclusion of PSI under this estimand;

More comprehensively discuss why other plausible confounders (e.g., SES, baseline physical activity, cognitive function) were excluded;

If model complexity was a constraint, please report the sample size and modeling approach used to evaluate overfitting.

[Response]

We sincerely appreciate the reviewer’s insightful comments. “PSI” is not considered to be associated with pet ownership; however, it is regarded as a strong prognostic factor that significantly influences the outcome. The primary objective of our study was to estimate the controlled direct effect of pet ownership on the outcome, and thus we included “PSI” as a covariate in the analysis. Although the results remained similar, we also conducted a sensitivity analysis excluding “PSI” to evaluate the total effect, which is presented in S1 Table. “Age” and “sex” were considered confounding factors. “Dementia” was also included in the analysis as a confounder, as it may be influenced by “socioeconomic status” (which affects pet ownership) and may also impact the outcome. While we acknowledge that variables such as “socioeconomic status”, “baseline physical activity” and “psychological status” could potentially serve as important covariates, we were unable to include them in the analysis due to a lack of consistent data. Our analysis was based on data carefully extracted from electronic medical records, and variables not recorded in the records could not be incorporated. We recognize this as a limitation of our study. However, these were not consistently recorded in the electronic medical records and could not be incorporated into the model in a reliable and systematic manner.

We have clarified these points in the revised Methods section (Lines 131-147) and Discussion section (Lines 254-285).

Reviewer: 3

Lines 139-140: you mentioned that Pneumonia severity may act as a mediator and a previous statement is that Pet ownership is unlikely to influence the severity of Pneumonia. A mediator should be in the middle of the pathway between ethe exposure and the outcome. If Pneumonia was not caused by pet ownership it cant be a mediator. However, you can still adjust for it if it causes the outcome; so no need to change the analysis.

[Response]

Thank you very much for your insightful comment regarding the role of PSI in our causal framework. In our study, we assumed that pet ownership and pneumonia severity (as measured by the Pneumonia Severity Index, PSI) are not causally related. However, because PSI is a strong prognostic factor for functional outcomes such as the Barthel Index (BI), we included it in the model as a precision variable to improve the accuracy of the estimated association. Our analysis was intended to estimate the direct effect of pet ownership on physical function, not the total effect that may be mediated through pneumonia severity. However, to also examine the total effect of pet ownership, we have added the corresponding results in S1 Table. To reflect this more clearly, we revised Lines 138-141 as follows: ”Although pneumonia severity, as reflected by PSI, is unlikely to be affected by pet ownership, it may significantly influence physical function outcomes. Pneumonia severity index may act as a prognostic factor, so we included the PSI as an adjustment variable to examine the direct effect of pet ownership on physical function.”

Fig1: any Inclusion or exclusion that happens on future events after the exposure can cause "Immortal time Bias"; lacking records of pet ownership and being hospitalized for the first time happens at the admission. Records missing should be treated as data missing not as an exclusion criteria. Patients that died during hospitalization should still be included and then censored at death. Please update the graph and the analysis accordingly. The graph should have no conditioning on future events (i.e. no inclusion, or exclusion criteria on the follow up period), and the characteristics table (the analysis) should include patients who owns pets upon admission, later on, if death occurs, they may be censored. Meaning they should be added to the baseline variables calculations, and then censored at the outcome.

[Response]

Thank you very much for your insightful feedback. After re-examining our approach, we fully agree with your assessment. In response, we revised our analysis to include all patients regardless of missing pet information, missing rehabilitation records, or in-hospital death, as these should not be treated as exclusion criteria. Consequently, we revised the Fig.1 accordingly, and updates to Tables 1

Line 84-86 ” We included patients aged ≥65 years who were hospitalized for pneumonia and whose medical records at the time of admission contained terms related to pet ownership.”

---

## [Decision Letter · Decision Letter 2]

31 Jul 2025

Association between Pet Ownership and Physical Function at Discharge in Hospitalized Older Adults: A Retrospective observational study

PONE-D-25-10406R2

Dear Dr. Ishihara,

We’re pleased to inform you that your manuscript has been judged scientifically suitable for publication and will be formally accepted for publication once it meets all outstanding technical requirements.

Kind regards,

Amr Ehab El-Qushayri

Academic Editor

PLOS ONE

Reviewers' comments:

Reviewer's Responses to Questions

**Comments to the Author**

Reviewer #2: All comments have been addressed

Reviewer #3: All comments have been addressed

2. Is the manuscript technically sound, and do the data support the conclusions?

Reviewer #2: Yes

Reviewer #3: Yes

3. Has the statistical analysis been performed appropriately and rigorously?

Reviewer #2: Yes

Reviewer #3: Yes

4. Have the authors made all data underlying the findings in their manuscript fully available?

Reviewer #2: Yes

Reviewer #3: Yes

5. Is the manuscript presented in an intelligible fashion and written in standard English?

Reviewer #2: Yes

Reviewer #3: Yes

Reviewer #2: (No Response)

Reviewer #3: (No Response)

**Do you want your identity to be public for this peer review?** For information about this choice, including consent withdrawal, please see our Privacy Policy

Reviewer #2: No

Reviewer #3: No

---

## [Editor Report · Acceptance letter]

PONE-D-25-10406R2

PLOS ONE

Dear Dr. Ishihara,

I'm pleased to inform you that your manuscript has been deemed suitable for publication in PLOS ONE. Congratulations! Your manuscript is now being handed over to our production team.

Kind regards,

on behalf of

Dr. Amr Ehab El-Qushayri

Academic Editor

PLOS ONE